# Overview of Research on Digital Image Denoising Methods

**DOI:** 10.3390/s25082615

**Published:** 2025-04-20

**Authors:** Jing Mao, Lianming Sun, Jie Chen, Shunyuan Yu

**Affiliations:** 1Graduate School of Environmental Engineering, The University of Kitakyushu, Kitakyushu 808-0135, Japan; 2Department of Information Systems Engineering, The University of Kitakyushu, Kitakyushu 808-0135, Japan; sun@kitakyu-u.ac.jp; 3School of Electronic and Information Engineering, Ankang University, Ankang 725000, China

**Keywords:** image denoising, BM3D, neural network, deep learning

## Abstract

During image collection, images are often polluted by noise because of imaging conditions and equipment limitations. Images are also disturbed by external noise during compression and transmission, which adversely affects consequent processing, like image segmentation, target recognition, and text detection. A two-dimensional amplitude image is one of the most common image categories, which is widely used in people’s daily life and work. Research on this kind of image-denoising algorithm is a hotspot in the field of image denoising. Conventional denoising methods mainly use the nonlocal self-similarity of images and sparser representatives in the converted domain for image denoising. In particular, the three-dimensional block matching filtering (BM3D) algorithm not only effectively removes the image noise but also better retains the detailed information in the image. As artificial intelligence develops, the deep learning-based image-denoising method has become an important research direction. This review provides a general overview and comparison of traditional image-denoising methods and deep neural network-based image-denoising methods. First, the essential framework of classic traditional denoising and deep neural network denoising approaches is presented, and the denoising approaches are classified and summarized. Then, existing denoising methods are compared with quantitative and qualitative analyses on a public denoising dataset. Finally, we point out some potential challenges and directions for future research in the field of image denoising. This review can help researchers clearly understand the differences between various image-denoising algorithms, which not only helps them to choose suitable algorithms or improve and innovate on this basis but also provides research ideas and directions for subsequent research in this field.

## 1. Introduction

Images are among the main ways for humans to acquire and exchange messages. With the popularization of digital imaging devices, the range of image applications has expanded, and the types of images have become more abundant. Two-dimensional amplitude-type images are the most common image category. By recording the intensity of light, it can show the form, texture, and brightness of objects in two-dimensional form, such as daily photographs, which are intuitive presentations of different natural scenes. Hyperspectral images add a spectral dimension based on two-dimensional space, which can obtain continuous spectral band information. They have been widely used in remote sensing, medicine, agronomy, and other fields. Spatio-temporal images are varied in both time and space, such as satellite remote sensing images and video streaming. Layered images can clearly show the internal structure of objects and are commonly used in medical diagnosis, industrial non-destructive testing, and other fields. Terahertz images use imaging of the interaction of terahertz waves with matter, which have great potential in security inspection and biomedical imaging. Digital holograms record the amplitude and phase information of an object’s light waves through the interference and diffraction of light and realize the reproduction of the object’s three-dimensional information through the digitized processing and reconstruction of the interference fringes. Holograms are commonly used in fields such as microscopic particle observation, imaging of biological samples, and measurement of three-dimensional objects. These different types of images play a key role in their respective fields and jointly promote the development and progress of science and technology. However, images are unavoidably polluted by noise in acquisition and transmission, which reduces the image quality as a result of outside elements, acquisition devices, and storage media. Damaged images are detrimental to the transmission of information and cause an incalculable impact on the following image processes: feature extraction, text detection, and image segmentation [1]. Accordingly, image-denoising techniques have become a research topic in image processing, computer vision, and related applications.

Researchers have developed corresponding denoising techniques and methods for the unique characteristics of different types of images. Katkovnik et al. proposed a sparse phase imaging method based on the complex-domain nonlocal BM3D technique, which was innovative and applied to deal with the problems related to the denoising of hyperspectral images [2,3]. Dutta et al. [4] proposed the application of deep learning to terahertz image denoising for the non-destructive analysis of historical documents, using deep learning to automatically learn complex features and patterns from the data to solve the noise problem in terahertz images. Wu et al. [5] proposed an unsupervised deep residual sparse attention network for retinal optical coherence tomography (OCT) image denoising. Bianco et al. [6] proposed a framework combining multi-hologram coding, block grouping, and collaborative filtering to realize quasi-noise-free digital holographic reconstruction. Two-dimensional amplitude-type images, especially natural images, are closely connected with people’s daily life and have an extremely wide range of applications, so the denoising problem of such images attracts a lot of attention from the academic community and is also the main research object of this review.

Two-dimensional amplitude-type image-denoising methods have a long history, and the earliest methods were mainly filter-based denoising methods, which were performed in the spatial domain or transform domain. Typical spatial domain methods include Gaussian filtering [7], Wiener filtering [8], bilateral filtering [9], and so on. Over recent years, as computer technology has developed, image-denoising methods have been continuously improved, and the most typical denoising method is the well-known BM3D framework [10], which combines the nonlocal similarity features of natural images and the sparse presentation in the transformation domain. At an early stage, most of the work associated with denoising images concerns the filtering of single-channel grayscale images. In recent years, advances in imaging systems and techniques have greatly expanded the information saved and displayed in color images, which could reproduce real scenes more realistically. Over the last two decades, representative BM3D methods have been significantly applied to multidimensional images in two distinct manners. The first approach was to use some kind of relevant transformation, such that every channel in the transformation space could be filtered individually via some efficient one-channel denoiser. For example, the CBM3D method [11] created a YCbCr color transform for natural RGB images, which provided almost-optimal solution correlation for color data. The other solution was to use channel or band correlation to model the entire multidimensional image dataset by jointly processing it. Maggioni et al. [12] extended BM3D to sparse 4D transform-domain collaborative filtering (BM4D) by using 3D pixel cubes for color image denoising. Although traditional methods have made significant achievements in image denoising, they also have many drawbacks. In the past few years, with the development and application of neural network technology, they have achieved remarkable results in the field of image denoising.

This review aims to make a comprehensive and systematic summary and comparative analysis between traditional image-denoising methods and deep neural network-based image-denoising methods. The main work is as follows:

(1) Analyzed the basic frameworks of classical traditional denoising methods and deep neural network-based denoising methods, and classified various denoising methods according to their principles, characteristics, and other factors, to lay the foundation for subsequent research.

(2) Relying on a public denoising dataset, a rigorous and comprehensive comparative evaluation of existing denoising methods was carried out from both quantitative analysis and qualitative analysis.

(3) By reflecting on the current status of the image-denoising field, we point out the hidden challenges and also look forward to the future research direction, providing a guideline for subsequent scientific research work.

## 2. Image-Denoising Framework

Image noise mainly comes from the process of image acquisition and transmission. For example, in the process of image acquisition, a light-sensitive device is affected by the brightness of the light and the different ambient temperatures, which would affect the imaging effect of the image and generate noise. In the process of image transmission, noise was generated due to the imperfection of the sending and receiving equipment and the poor transmission channel. According to the distribution law of noise, it can be divided into Gaussian noise, Poisson noise, and Skellam noise.

(1) Gaussian Noise

Gaussian noise is the most common type of noise in imaging and is represented by a Gaussian distribution function [13]. It is additive and independent. For example, when shooting with a camera, the amplifier inside the camera introduces noise when amplifying the signal. In conventional film photography, the uneven distribution of silver salt particles in the film causes this noise.

An image containing Gaussian noise can be represented as $g\left(x,y\right)=f\left(x,y\right)+n\left(x,y\right)$, where g is the noise-containing image, f is the original image, and n is the additive noise on the pixels.

(2) Poissonian Noise

When an image is affected by photon noise, its noise presents a Poissonian distribution [14]. Photon noise originates from the image acquisition process, with the random fluctuation in the photon impacting the sensor. In addition to photon noise, factors such as deviations in the calibration process of the machine, interference in the data transmission chain, and the characteristics of the storage medium also produce noise that similarly obeys the Poisson distribution.

(3) Skellam Noise

The noise generated during terahertz pulsed time-domain holographic raster scanning based on a balanced detection system and low-photon imaging such as X-ray fluoroscopy can be described by the Skellam distribution [15]. Skellam distribution noise variance is independent of the true signal and exists in imaging scenarios involving the random behavior of photons or similar independent Poisson processes.

Whatever the type of noise is, it degrades the quality of an image and interferes with the subsequent image analysis and processing. Therefore, before further processing of the image, it has to be denoised in the first place. In this review, the focus is on the techniques and methods of Gaussian noise removal.

### 2.1. Traditional Denoising Methods

Traditional denoising methods fall into three main types.

(1)Noise removal using filter techniques

It is mainly performed in the space and transform domains. The basic principle is to process each pixel point in an image and use the information of its neighboring pixel points to correct the pixels at that point, thus smoothing the image. Take the classical bilateral filtering algorithm as an example; it is an adaptive weighted filtering method. The pixel values of the output image are weighted combinations of neighboring pixels. The weights are determined based on the geometric distance and gray value differences between the filtered point and the neighboring pixels. In other words, the weights are dynamically adjusted with the geometric distance of the pixels and the similarity of the pixel gray values. The pixel point that is closer to the filtered point and has a smaller difference in the gray level is assigned a larger weight, and vice versa, a smaller weight. Bilateral filtering has significant advantages in edge protection and image smoothing compared to Gaussian filters and median filters. The disadvantages of the bilateral filtering algorithm are mainly the higher computational complexity and slower processing speed because it requires complex weight calculation for the neighborhood of each pixel. In addition, the parameter selection is more difficult, and it needs to be adjusted according to the characteristics of different images; otherwise, it might affect the filtering effect. Based on the theoretical foundation of the classical algorithm, scholars have further explored the classical denoising algorithm in the spatial and transform domains. Tang et al. [16] proposed a modified curvature filtering algorithm. The algorithm combined the half-window triangular tangent plane and the minimum triangular tangent plane, replaced the traditional minimum triangular tangent plane projection operator with the projection operator, and amended the regular energy function to add the regular energy of local variance according to the characteristics of the strong noise image that existed in the strong noise speckle, which improved the denoising performance of the algorithm. Although the algorithm had a good denoising effect on strong noise, it cannot be tuned adaptively to the tangent plane projective operator in the neighborhood. It also took a long time to run. Cheng et al. [17] designed a remote sensing image denoising method based on a curvilinear waveform transform and goodness-of-fit test. The algorithm was designed to normalize the curvilinear coefficients in the curvilinear domain and to perform a local test on the normalized curvilinear coefficients using the goodness-of-fit test (GOF), which removed the noise coefficients and left the signal coefficients. Finally, the inverse normalized signal curvilinear coefficients were inversely curvilinear transformed to obtain the denoised remote image. Zhang et al. [18] accomplished image denoising by setting appropriate tuning arguments, selecting fixed thresholds statistically, and adding a tuning parameter for reducing the deviations from a constant between original wavelet coefficients and estimated wavelet coefficients.

(2)Denoising using sparse coding

The core idea of sparse coding denoising is to utilize the sparsity of the signal to sparsely represent the noise-containing signal in an overcomplete dictionary, and denoising is achieved by removing the non-sparse portion of the noise in the dictionary. The K-SVD algorithm is a typical representative of this [19]. The principle is divided into three main steps. The first step is dictionary learning. The algorithm first randomly initializes an overcomplete dictionary D. Then, for a given collection of noisy image blocks, an optimization problem is solved to find the sparse representation coefficient X of each image block on the dictionary, such that DX approximates as closely as possible to the original image block, while X is sparse, i.e., most of the elements are zero. The second step is the dictionary update. After obtaining the sparse representation coefficients, the algorithm updates the dictionary using singular value decomposition (SVD) methods to better fit the features of the image blocks. The third step is the denoising process. After several iterations of dictionary learning and updating, a dictionary that can represent the image block well is obtained. The new noise-containing image is divided into multiple image blocks, and then each image block is sparsely represented using the learned dictionary, and, finally, the denoised image is reconstructed by these sparse representation coefficients and the dictionary. The K-SVD algorithm can effectively utilize the sparsity of the image for denoising and better retain the details and texture information of the image while removing the noise. However, it also has drawbacks, including high computational complexity, resulting in slow operation speed. It is more sensitive to the initialization of the dictionary, and different initializations may affect the final denoising effect. Therefore, researchers have continuously improved this algorithm. Li et al. [20] proposed an image-denoising algorithm based on adaptive match tracking. The algorithm first solved the sparse coefficient problem through the adaptive matching tracking mechanism. Then, the dictionary was trained into an adaptive dictionary that can effectively reflect the structural features of the image using the K singular value decomposition algorithm. Finally, the sparse coefficients were integrated with the adaptive dictionary to reconstruct the image. Li et al. [21] proposed an image-denoising method for a 2D multipath matching tracking algorithm. The approach divided the noisy image into image blocks and used the dictionary that trained on the noisy image to sparsely represent each image block using breadth-first and/or depth-first 2D multipath matching tracking algorithms, which removed the noise from every image block, then reconstructed the denoised image block into a complete denoised image. Yuan et al. [22] used similar block matching globally to obtain sparse coefficient estimates for ideal images. The class dictionary and estimated sparse coefficients were utilized to implement image denoising.

(3)Use of external prior for denoising, also known as model-based denoising

From a Bayesian perspective, this class of methods defines the denoising task as an optimizing issue based on maximum a posteriori probability, in which prior knowledge plays a crucial role in the optimization task. There are various classical models, mainly including nonlocal self-similar models [23,24,25,26], gradient models [27,28,29], and Markov random field models [30]. There are many representative approaches such as NLM [23] and BM3D [10]. They had a similar idea of combining filtering with a self-similarity model to search for familiar regions in the whole image in small image blocks and performing an averaging operation to remove the noise.

BM3D is a three-dimensional transform-domain filtering-based algorithm, which is among the best image-denoising techniques available. Many efficient noise removal approaches have been introduced based on the BM3D algorithm. The algorithm is organized into two main steps. The first step is base estimation. The image is divided into sub-modules of fixed size, and each block in the image is estimated block by block. The blocks are grouped by their similarity to each other, and these blocks are aggregated into a three-dimensional array. Afterward, the 3D transformation is performed on the 3D array. In the end, the base estimation of the image is obtained by weighting the blocks with overlap through aggregation. The second step is to perform a second estimation of each block using the base estimation image obtained in the first step. The block matching is performed again to find the location of the block that is similar in the benchmark estimation image. After matching, two 3D arrays are obtained. Joint Wiener filtering is performed on the two 3D arrays formed. In the end, the weighted average of the estimated values of the overlapping blocks is used to obtain the ultimate denoised image. A flowchart of the BM3D algorithm is given in Figure 1.

Other typical methods include weighted nuclear norm minimization (WNNM) [31]. Liu et al. [32] suggested a novel image-denoising approach by using the coefficient matrix in the low-rank representation model to impose a total variation paradigm constraint. Lv et al. [33] incorporated relative total variation (RTV) into weighted nuclear norm minimization (WNNM), imposed RTV norm constraints on the WNNM low-rank model, and proposed a relative total variation and weighted nuclear norm minimization (RTV-WNNM) image-denoising approach. Model-based approaches can initially achieve noise removal, but most of them have two obvious drawbacks. The first one is the need to select the parameters manually, and the second one is the complex optimization process involved in the testing phase, which requires long processing time. More traditional denoising algorithms are shown in Table 1 for detail.

Through the study of the above literature, the advantages and disadvantages of various types of algorithms are summarized as follows:

The advantages of filter-based denoising algorithms mainly include the following: (1) Higher computational efficiency compared to the other two types. It is usually based on a simple convolutional operation, which can quickly process the image. (2) The principle of the algorithm is relatively simple, the implementation difficulty is low, the hardware requirements are not high, and it can be run on devices with limited resources. (3) For Gaussian noise and other noise with specific statistical characteristics, a good denoising effect can be achieved by designing a suitable filter. The disadvantages mainly include the following: (1) Image blurring. Although the Gaussian noise can be suppressed to a certain extent, when smoothing the noise, the image details and edge information are smoothed together, resulting in an overall blurring of the image. (2) Different filtering algorithms are designed for specific noises, and the effect of changing the type of noise may become worse. (3) The parameter settings are poorly self-adaptive. With different parameters, the filtering effect varies greatly, and there are no general optimal parameters. Given the advantages and disadvantages of such algorithms, it can be concluded that this class of algorithms is suitable for denoising simple textured images with low noise levels, relatively high real-time requirements, and simple textures.

The advantages of image-denoising algorithms based on sparse coding mainly include the following: (1) Good detail retention. It can remove noise while retaining the detailed information of the image better so that the denoised image is clearer. (2) By learning the sparse representation of the image, it can adaptively denoise according to the local features of the image and has good adaptability to different types and distributions of images. Disadvantages mainly include the following: (1) High computational complexity. The algorithm involves a large number of matrix operations and optimization solutions, high computational cost, and slow processing speed; when the image resolution is high or the amount of data is large, it is especially more obvious. (2) Dictionary learning problem. The dictionary learning process is time-consuming and depends on a large amount of training data. If the training data differ greatly from the actual application image, the denoising effect is affected. (3) Difficulty of parameter adjustment. Parameters need to be adjusted according to the image and noise characteristics. There is no universal optimal value, and repeated experiments and debugging are required, making it more difficult to use. This type of algorithm is suitable for processing images with rich texture and details, moderate noise levels, and the need for high-precision denoising.

The advantages of image-denoising algorithms based on external a priori assumptions mainly include the following: (1) Preservation of image features and strong detail recovery ability. When the external a priori knowledge accurately reflects the true characteristics of the image, the algorithm is able to recover the details in the image with high accuracy based on this knowledge. (2) High flexibility. The appropriate external prior knowledge can be selected according to different application scenarios and needs, with high flexibility and customizability. Disadvantages include the following: (1) Low computational efficiency. The testing phase often involves complex optimization problems, large computational volume, and time-consuming denoising, and it is difficult to take into account high performance and computational efficiency. (2) Hyper-parameterization. The model is mostly non-convex, so the parameters have to be selected manually. Different parameters have a big impact on the denoising performance, and the tuning of parameters depends on experience and is cumbersome. (3) Strong dependence on a priori assumptions. When the a priori assumption is inaccurate or inapplicable, it would misjudge the image region, destroy the details of the image structure, fail to effectively distinguish between the noise and the signal, and may also introduce artifacts. Synthesizing the advantages and disadvantages of such algorithms, it can be concluded that this class of algorithms is suitable for processing images with specific a priori knowledge, such as medical images. Using the existing prior knowledge of medical images can accurately remove noise and retain key medical features. This type of algorithm is also suitable for processing high-precision images in specialized fields, such as satellite remote sensing images, aerial photography images, and so on. These images usually require high-precision denoising processing, and prior knowledge in specialized fields could be utilized to improve the accuracy and reliability of denoising.

### 2.2. Deep Learning Denoising Approaches

According to the difference in the type of noise that the model can handle, deep learning denoising algorithms are categorized into four groups: additive Gaussian white noise image denoising, real noise image denoising, blind noise image denoising, and mixed noise image denoising.

(1)Denoising approaches for additive Gaussian white noise images

Additive Gaussian white noise is prevalent in various imaging systems and communication channels with explicit mathematical models and statistical properties, which facilitates theoretical analysis and algorithm design, based on which many classical denoising algorithms were developed. Mao et al. [53] proposed a deep all-convolutional encoding–decoding framework, which introduced symmetric jump connections between convolutional and anti-convolutional layers and was successfully used for image reproduction tasks like denoising and super-resolution. Tai et al. [54] created a long-term memory network (MemNet) that was 80 layers deep. Memory modules were constructed by using recursive units and gating units to determine the proportion of the current short-term memory and pre-learned features in the subsequent information transfer. MemNet had a strong learning ability and had a good gain. However, it was too computationally intensive and took a long time to process. Zhang et al. [55] proposed a denoising convolutional neural network (DnCNN) that combined batch normalization and residual learning techniques. Although the denoising approach achieved exceptionally outstanding results, the entire system required too much iteration to achieve a better model. The speed and vergence of the overall algorithm were not prominent enough. Zhang et al. [56] presented a fast and flexible denoising network (FFDNet). FFDNet was based on the network structure of DnCNN. It was able to process additive Gaussian white noise (AWGN) with a noise intensity in the range [0, 75] using only one training model. However, it must choose the corresponding noise level mapping with the noisy image together with the input network for training, and the model training complexity is relatively high. Zhang et al. [57] designed a multiscale feature learning convolutional neural network (MSFLNet). It was composed of three feature learning (FL) modules, a reconstruction generation (RG) module, and a residual connection. It can effectively study the characteristic information of the image and increase the noise removal efficiency. Valsesia et al. [58] proposed a graph-convolutional-based operation to create a nonlocal sensory field. A graph-convolutional image-denoising (GCDN) model for the efficient representation of self-similarity was obtained by dynamically computing the similarity of graphs in hidden features. However, the generalization ability of the model could be improved. Table 2 shows more information about Gaussian white noise image-denoising approaches based on deep learning.

(2)Denoising methods for real noisy images

Image acquisition is affected by a variety of factors, and the noise situation is also complex and varied. The study of real image-denoising algorithms can make the image maintain better quality in different scenarios. Yan et al. [76] derived a noise map from the noise image directly, thereby realizing the unsupervised noisy modeling and accomplishing the denoising of unpaired real noisy images. The network architecture was self-consistent generative adversarial networks (SCGANs). Zhao et al. [77] proposed end-to-end denoising of dark burst images using recurrent fully convolutional networks. The original burst image was directly mapped to the SRGB output to generate the optimal image or to generate a multi-frame denoised image sequence with a recurrent fully convolutional network (RFCN). Although this denoising framework is highly flexible, it does not extend its framework to video denoising, and the framework is not yet portable. Abuya et al. [78] used an integrated approach to eliminate noise from the CT image disturbed by adductive Gaussian noise, which integrated an anisotropic Gaussian filter and wavelet transform with a deep learning denoising convolutional neural network and demonstrated excellent performance in maintaining image quality and preserving fine details. Gou et al. [79] proposed a multiscale adaptive network (MSANet). This network simultaneously considered the scale characteristics and cross-scale complementarity and integrated them into a multiscale design, which effectively improved the denoising performance of images. However, the algorithm still does not consider the loss of image details. Bao et al. [80] designed an MCU-Net denoising method, which added a branch based on a residual dense block of empty spatial pyramid pools (ASPPs). These model sub-paths extracted image features of different resolutions, which were fused at the end of the network for denoising. Table 3 shows more information about deep learning-based denoising approaches for realistic noisy images.

(3)Denoising methods for blind noise images

Blind noise image denoising refers to the processing of noise-contaminated images to restore their original clarity when the type, intensity, and distribution of the noise are unknown. Yang et al. [91] designed a new approach to evaluate the noise level from a single image with multi-column convolutional neural networks. However, this algorithm has not been implemented to denoise natural images. Guo et al. [92] continued the idea of FFDNet. A two-stage network (CBDNet) was designed from the consideration of a noise level map. Firstly, the noise level map was obtained through the noise estimation sub-network, and then it was fed into the non-blind denoising sub-network together with the noisy image, to achieve a certain degree of image blind denoising. Tao et al. [93] proposed a residual dense attentional similarity network (RDASNet) for image denoising. The network extracted the partial features of the image via CNN and attended to the overall message of the image via the attentional similarity module (ASM). In addition, it used inflated convolution to expand the sensory field to pay greater attention to more global features. Table 4 shows more information about the blind noise image-denoising approaches based on deep learning.

(4)Denoising methods for mixed noise images

It is a technology for denoising images, which are disturbed by many different types of noise at the same time. Zhang et al. [100] proposed a three-layer hyper-resolution network for multiple degeneracy, which was a general framework with a dimensionality expansion strategy to handle multiple and even spatially varying degradations. Li et al. [101] designed a self-supervised two-phased denoising approach for EBAPS image mixture noise. Self-supervised learning was achieved by combining UNet [102] and BSN and drawing on the iterative training idea of IDR [103]. Table 5 shows more information about the deep learning-based denoising approaches for mixed noise images.

In 1980, Sullivan et al. [108] as well as Zhou et al. [109] first used deep neural networks for image denoising. Compared with the traditional typical image denoising approach, deep convolutional neural networks show powerful machine learning capabilities. Using many noise-containing image sample data for training can effectively improve the ability of a network to fit various intensities of noise and make it have a stronger generalization ability. The CNN-based image-denoising approach usually adopts the strategy of learning clear images. Among many algorithms, DnCNN is undoubtedly the more efficient approach, combining residual learning and batch-normalized optimization training for image denoising. The DnCNN model emphasizes the complementary roles of residual learning and batch normalization in image recovery. It also achieves fast convergence and denoising capabilities despite a deeper network. So far, this algorithm is one of the more advanced denoising algorithms in the field of image denoising, and it is one of the more prominent denoising algorithms. The network is mainly divided into a preprocessing layer, an intermediate processing layer, and an image reconstruction layer. The preprocessing layer applies 64 filters of size 3 × 3 for the initial feature extraction of noisy images to produce 64 feature maps. Then, the activation function ReLu is used for nonlinearity. The intermediate processing layer performs the deep feature extraction of the image, corresponding to the 2 ~(D-1) layers of the neural network. Every layer uses 64 filters of size 3 × 3 × 64 with batch normalization added in between convolution and ReLU. The image reconstruction layer using C filters of size 3 × 3 × 64 is used as the reconstructed output.

Although the DNCNN model has greatly improved the denoising performance, there is still much room for improvement with the continuous development of deep learning technology. Many researchers have improved it with new techniques. Based on the above deep learning-based denoising methods, it can be seen that the improvement direction of the DNCNN model can be divided into three aspects:

First, optimization of the network structure. By deepening or broadening the network, the expressive power of the model can be significantly improved, enabling it to learn more complex image features [65,69,70,72,73,90]. In this process, it is necessary to pay attention to the phenomenon of over-fitting and the resulting high computational complexity. Therefore, the introduction of a residual connection is also one of the optimization ideas [61,65,71,81]. Residual connection can effectively solve the problem of gradient disappearance, which not only makes the network training process smoother but also retains the image details when denoising better. In addition, multiscale convolution and convolution operations with convolution kernels of different sizes can extract image features at different scales, thereby enhancing the model’s ability to capture image details and structure [61,63,65,73]. The convolutional neural network was optimized using the above method so that the noise reduction model based on the convolutional neural network (CNN) can accurately learn the difference between noise and image features. Removing white Gaussian noise can not only effectively reduce the noise level but also better retain the edge and texture details of the image.

Second, introducing an attention mechanism. The introduction of the attention mechanism opens a new path for model optimization [64,67,87,90,91,92,93]. Among them, the channel attention mechanism allows the model to automatically learn the importance of different channels, pay more attention to important channels, suppress those unimportant channels, and improve the feature extraction ability of the model. The spatial attention mechanism enables the model to accurately focus on important spatial regions in the image, enhance the ability to capture local features of the image, better retain the details and edges of the image in the process of denoising, and significantly improve the quality of the image after denoising. The deep learning model combined with the attention mechanism can adaptively learn and estimate the unknown noise characteristics to achieve more effective denoising of blind noise images.

Third, enhancing data processing. In the data processing phase, a variety of data enhancements can be performed. For example, rotating, flipping, scaling and other operations on training data can greatly enrich the diversity of data, effectively improve the generalization ability of the model, and enable it to perform well in the face of various complex noises [85,92]. At the same time, the adversarial training mechanism was introduced to make the generator and discriminator progress together in the adversarial game so that the generator can generate more realistic denoised images and further improve the overall performance of the model [82,83,96,104,107]. The application of these methods has greatly improved the denoising performance of deep convolutional neural networks in real noise images and mixed noise images.

## 3. Datasets

Most effective deep learning image-denoising methods mainly use paired data training. The training data are required to be (noisy, clear) image pairs. There are currently three ways to construct (noisy, clear) image pairs, summarized as follows:

The first is to obtain clear and high-quality images from image databases, such as the Berkeley Segmentation Dataset [110], Waterloo Exploration Database [111], and DIV2K [112]. Then, add noise, such as Gaussian noise, Poisson noise, and so on, to the clear image to synthesize the image with noise, and build the paired training data. This way of synthesizing noise images is relatively simple, but there are some differences with real noise images.

In the second method, a low-ISO image is taken as a clear image, and the corresponding high-ISO image is taken as a noisy image in the same scene. This method only uses a single low-ISO image as a clear image, which may carry noise, or there are problems with brightness, contrast, and other conditions that do not correspond to the noisy image.

The third is to shoot multiple images for the same scene, and then carry out operations such as image alignment and post-processing to generate almost noise-free images by weighted average. This method uses multiple images to obtain clear images with relatively high quality but requires a large amount of data as the premise, a large amount of work to shoot images, and the calibration rules between multiple images are more stringent.

Most researchers use the first method to create training image pairs. Due to the complex sources of noise, there may be different types of noise in real noisy images. If it was assumed to be the sum of independent random variables with different probability distributions, according to the central limit theorem, the more noise of different properties, the more the normalized result would approximate the Gaussian distribution [113]. Therefore, the Gaussian noise that was closest to the real noise was usually chosen for the experiment. The detailed information of the synthetic noise image dataset is shown in Table 6.

Considering that real noise is more complex, some researchers have applied the second and third methods to construct real (noisy, clear) image pairs as training or test data. The commonly used datasets are shown in Table 7. These images were collected with different cameras and different ISO settings.

## 4. Evaluation Standards

A noisy image is denoised to obtain an image that theoretically removes the noise efficiently and retains the important information. This can be illustrated from the following aspects [118]: (1) the removal of noise; (2) the protection of edge texture and other details; (3) the degree of regional smoothing. Therefore, to evaluate the image quality after denoising, the above three aspects should be considered. At present, image quality assessment metrics can be categorized into subjective and objective assessments [119].

(1) Subjective evaluation

Subjective evaluation refers to the assessment of the quality of the image through visual observation and subjective feelings. The usual approach for subjective evaluation is the comparative observation method. It can be assessed artificially with a comparison between the denoised image and the original image and also by comparing the denoised image with different algorithms. In the subjective evaluation criteria, the mean subjective score (MOS) was that the observer used the dual-stimulus continuous-quality grading method for the image to be evaluated concerning the original image. The image to be evaluated and the original image were alternately played for a certain period to the observer according to certain rules, and then a certain time interval was set aside for the observer to score after the playback. Finally, the average of all the given scores was taken as the score of the image to be evaluated.

(2) Objective evaluation

The objective evaluation was used to assess the image quality via the discrepancy between the original image and the processed image. The most commonly used objective evaluation criteria were mean square error (MSE), pear signal-to-noise ratio (PSNR), and structural similarity (SSIM), as shown in Equations (1)–(3).(1)$$MSE=\frac{1}{H\times W}\sum_{i=1}^{H}\sum_{j=1}^{W}{\left[I(i,j)-\tilde{I}(i,j)\right]}^{2}$$(2)$$PSNR=10\times {log}_{10}\left[\frac{2^{8}-1}{\frac{1}{H\times W}\sum_{i=1}^{H}\sum_{j=1}^{W}{\left[I(i,j)-\tilde{I}(i,j)\right]}^{2}}\right]$$(3)$$SSIM=\frac{\left(2\mu_{1}\mu_{2}+C_{1})(2\sigma_{12}+C_{2}\right)}{\left({\mu_{1}}^{2}+{\mu_{2}}^{2}+C_{1}\right)({\sigma_{1}}^{2}+{\sigma_{2}}^{2}+C_{2})}$$

(3) Spearman rank order correlation coefficient

The evaluation index was used to measure the correlation between the rank of two variables. In image quality evaluation, it can be used to evaluate the consistency between objective evaluation indicators and subjective evaluation results.

## 5. Experimental Result

To make a comparison among the denoising performance of traditional and deep neural networks, quantitative and qualitative evaluation experiments were conducted on Set12, BSD68, CBSD68, Kodak24, McMaster, DND, SIDD, polyU, and CC datasets. The experimental hardware environment was Windows 11. The CPU was 11th Gen Intel (R) Core (TM) i7-11700T. RAM: 32 GB. The software environment was Matlab2023B, python 3.9, PyTorch 1.1.0.

### 5.1. Grayscale White Gaussian Noise Image Denoising

For grey images containing Gaussian white noise, traditional denoising methods were compared with deep learning denoising methods. Four traditional approaches (NLM, BM3D, WNNM, and EPLL) and ten deep learning-based denoising methods were included. The 10 deep learning-based denoising methods can be divided into three categories: (1) the classical single model trained for each noise level (MLP, TNRD, DnCNN, DudeNet); (2) the classical blind denoising models based on CNN that can handle various noise levels (IRCNN, ECNDNet, ADNet, FFDNet); (3) nonlocal self-similar priors in traditional methods were integrated into deep learning denoising models (NLRN, RNAN). Set12 was used as the test dataset. Gaussian white noise of different intensities was added to the images in Set12. The noise levels were 15, 25, 50, respectively, and these images were denoised.

The average PSNR values of the various approaches on the Set12 dataset with noise levels of 15, 25, and 50 are shown in Table 8. The bold text refers to the optimal results. It is evident that the NLRN had the most effective PSNR. At different noise levels, the average PSNR increase in NLRN compared to the BM3D method was about 0.77 dB, 0.81 dB, and 0.88 dB. Other deep learning denoising methods also had good denoising performance compared to traditional denoising methods, as shown in Figure 2. It was evident that the PSNR of deep learning denoising methods grew faster with the increase in noise intensity, indicating that deep denoising methods had better results in strong noise removal, showing the advantages of deep learning technology.

Figure 3 and Figure 4 show the denoising results of grayscale white noise images on the Set12 dataset using different methods when the noise levels were 15 and 23, respectively. It can be seen that among traditional algorithms, BM3D had excellent denoising performance, which can effectively remove the noise, and had excellent performance in preserving image details and texture. After denoising, the image had a clear visual effect and complete structure, as shown in Figure 3d and Figure 4d. WNNM had a good denoising effect. While suppressing noise, it can retain image edge and detail information well, especially when processing images with complex textures, as shown in Figure 3e and Figure 4e. EPLL can effectively remove noise and has certain advantages in recovering the high-frequency details of images, but it may produce artifacts, as shown in Figure 3f and Figure 4f. The effect of the deep learning denoising method was better than that of the traditional method. Compared with FFDNet and IRCNN, DudeNet had a better denoising effect, especially for images with rich textures, as shown in Figure 3i–k. NLRN combined nonlocal operations with recurrent neural networks. Nonlocal operations captured global context information by calculating the similarity between each pixel and all its neighborhood pixels and can handle small changes in complex scenes. With the introduction of recurrent neural networks, the model can gradually optimize the recovery results through several iterations and gradually approach the optimal solution in each iteration. As a result, the restored image edges and textures were sharper.

To verify the denoising efficiency of different algorithms, five grayscale images with a size of 512 × 512 were selected from the Set12 dataset to test in a single CPU environment. Table 9 shows the average time of processing a single image using the above different algorithms. As can be seen from Table 9, in traditional denoising algorithms, NLM had low efficiency due to excessive similarity calculation; BM3D had more matrix operations and data processing, and its efficiency was general; WNNM was more computative than NLM, and the efficiency was very low. EPLL was not efficient because it needed to solve complex optimization problems. Combining the denoising effect and efficiency of all kinds of denoising algorithms, it can be concluded that BM3D was slightly lower than WNNM in denoising performance but far better than WNNM in efficiency. In the deep learning denoising algorithm, DudeNet showed a similar denoising performance as NLRN, but its efficiency was relatively low. NLRN had the best denoising performance, and the efficiency of denoising was also good.

### 5.2. Color Image Denoising

For color image denoising, the classical traditional denoising approach, CBM3D, and six deep learning methods were selected for comparative analysis. CBSD68, Kodak24, Set5, and McMaster datasets were used as test sets. Gaussian white noise of different intensities was added to these datasets, the variance in the noise was 15,25 and 50, and the denoising experiment was carried out on these added images.

Table 10 shows the denoising outcomes of different approaches on different datasets for color images. The bold text indicates the optimal results. Take the BSD68 dataset as an example. When the noise level was 25, compared with the traditional denoising algorithm CBM3D, the average PSNR of DNCNN, FFDNET, DSNet [120], BRDNet, RPCNN [121], and IRCNN was improved by 0.6 dB, 0.5 dB, 0.57 dB, 0.72 dB, 0.53 dB, and 0.45 dB. Therefore, the deep learning-based denoising methods show significant advantages. In particular, the BRDNet denoising approach achieved the best results on all four datasets. Figure 5 shows the visual results of several methods for an image from the Set5 dataset with a noise level of 25. Figure 6 shows the visual results of several methods for an image from the CBSD68 dataset with a noise level of 50. Figure 7 shows the visual results of several methods for an image from the McMaster dataset with a noise level of 50. It is clear that BRDNet could recover more details and textures than the other approaches.

To verify the denoising efficiency of different algorithms, five color images of size 280 × 280 in the Set5 dataset were tested. Table 11 shows the average time taken by several different algorithms to process a single image. It can be seen that, among these algorithms, FFDNET was efficient in denoising and can quickly denoise color images. DNCNN and the traditional algorithm CBM3D had higher processing efficiency. BRDNet’s efficiency was in the medium range. However, IRCNN had relatively low efficiency due to iterative calculation.

### 5.3. Real Image Denoising

To test the denoising performance of deep learning technology in real noise images, public datasets, such as DND, SIDD, PolyU, and CC, were selected to carry out experiments. Three traditional denoising methods were compared with eight deep neural network denoising methods. The PSNR and SSIM values for various denoising approaches on the DND dataset and the SIDD dataset are listed in Table 12. Bold represents the optimal results. As can be seen from Table 12, the deep learning denoising algorithm had significant advantages over traditional denoising algorithms in DND and SIDD datasets. Taking SIDD as an example, compared with traditional denoising algorithms (CBM3D, EPLL, and KSVD), the GMSNet and MPRNet algorithms can increase the average PSNR by about 14 dB and SSIM by about 0.34. On the DND dataset, GMSNet performed equally well, with a PSNR of about 5 dB higher and SSIM of about 0.11 higher than traditional denoising methods. Through testing on DND and SIDD datasets, the performance of deep neural networks in the field of image denoising was better than that of traditional denoising methods.

Table 13 shows the PSNR and SSIM values for different denoising approaches on the CC and PolyU datasets, with bold indicating the optimal results. It can be seen that the traditional denoising approaches CBM3D and NLH displayed very competitive performance compared to the deep neural network algorithms DNCNN, FFDNet, and MIRNet in both datasets, whereas the deep neural network methods did not always demonstrate superiority over the traditional denoisers, which was largely due to insufficient training data.

## 6. Challenges, Opportunities, and Future Directions

In the area of digital image denoising, machine learning has shown significant advantages with its powerful feature extraction and pattern recognition capabilities. For example, it can effectively deal with complex noise patterns, denoising and retaining image details with excellent results. However, many problems have been exposed in practical applications: (1) Model training highly relied on the quality and quantity of training data. A slight deviation or insufficiency of data will seriously affect the denoising effect. (2) It consumed a great deal of computing resources. The deep neural network model had a complex structure, and the training and inference process required a large amount of GPU computing resources and a long time, which constrains its applications for devices with resource constraints, like mobile devices and embedded systems. (3) The trained model lacks generality. The characteristics of different types of noise were very different, and the existing machine learning denoising algorithms are often designed for specific types of noise and lack versatility. When faced with mixed noise or unknown kinds of noise, the denoising results of the model will be much less effective.

With the continuous development and maturity of new AI technologies, the research on digital image-denoising algorithms will show new research trends. (1) Explore new neural network architectures, such as developing more efficient convolutional neural network variants or Transformer-derived architectures, optimizing the network structure, reducing the number of parameters, and decreasing the computational complexity to improve the denoising efficiency and performance of the model. The denoising effect can also be improved by designing a more reasonable attention mechanism so that the model focuses more on the key regions of the image. (2) By fusing multimodal data, such as depth information and spectral information, to provide richer image features, it helps the model to more accurately distinguish between noise and real image information, to achieve a better denoising effect. For example, in medical imaging, information from MRI and CT images is combined for denoising and diagnosis. (3) Reinforcement learning is introduced into the field of image denoising, which allows the model to learn the optimal denoising strategy through interaction with the environment. Reinforcement learning can dynamically adjust the parameters and steps of the denoising algorithm according to different image noise situations and application requirements, and it can achieve an adaptive denoising process. (4) Domain adaptive and transfer learning techniques can be used. The models trained in one domain are quickly applied to other domains, reducing the dependence on data from the new domain and the cost of retraining. For example, a denoising model trained on natural images migrates to the domain of medical images by fine-tuning the model parameters to adapt to the characteristics of medical images.

## 7. Conclusions

With the development of digital technology, image processing has been widely used in various fields and has become an extremely important and indispensable technology. Before image processing, the effective removal of noise generated in image acquisition, transmission, and other links was a prerequisite to ensure the accuracy and reliability of subsequent processing. This review carried out a comprehensive and in-depth exploration of the field of image denoising and made a detailed comparison, analysis, and systematic summary of traditional methods and deep learning methods. Firstly, the classical denoising algorithms for image denoising were described and compared in detail. For simple noise, they could quickly realize the suppression of noise and had the advantages of low calculation cost and easy implementation. However, for complex noise or high noise, the traditional algorithm was not good at processing. Then, this review focused on deep learning techniques for image noise reduction. The research showed that deep learning technology opened up a new idea and method for image noise reduction through its powerful feature extraction and model fitting capabilities. Deep convolutional neural networks can accurately learn the difference between noise and image features and can not only effectively reduce the noise level but also better preserve the edge and texture details of the image when removing white Gaussian noise. The deep learning model combined with the attention mechanism can adaptively learn and estimate the unknown noise characteristics to achieve more effective denoising of blind noise images. Generative adduction networks (GANs) show unique advantages in denoising real noise images and mixed noise images. To show the performance of different methods more directly, the denoising performance of different networks was verified and compared to the standard dataset. The evaluation indexes of deep learning networks such as PSNR and SSIM were better than those of traditional methods, which further advanced deep learning technology in the field of image denoising. Finally, it summarized the challenges, opportunities, and future development directions of deep learning denoising algorithms. At present, deep learning denoising algorithms face challenges, such as high model complexity, large computing resource requirements, and dependence on large-scale high-quality datasets. However, with the rapid development of hardware technology and the continuous emergence of new algorithms, future deep learning denoising algorithms will be more intelligent, adaptive, and deeply integrated with traditional methods. This review provides a valuable reference for those seeking to apply digital image-denoising technology in a specific field or to promote the development of this field with innovative algorithm design.

## Figures and Tables

**Figure 1 sensors-25-02615-f001:**
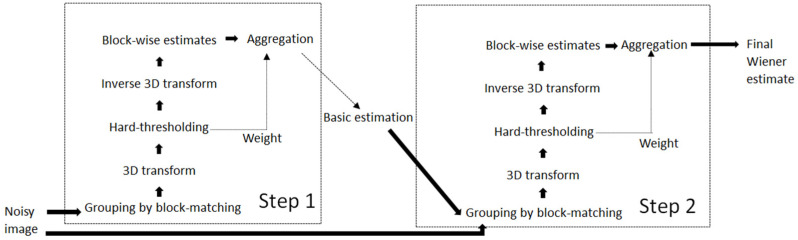
BM3D algorithm flowchart.

**Figure 2 sensors-25-02615-f002:**
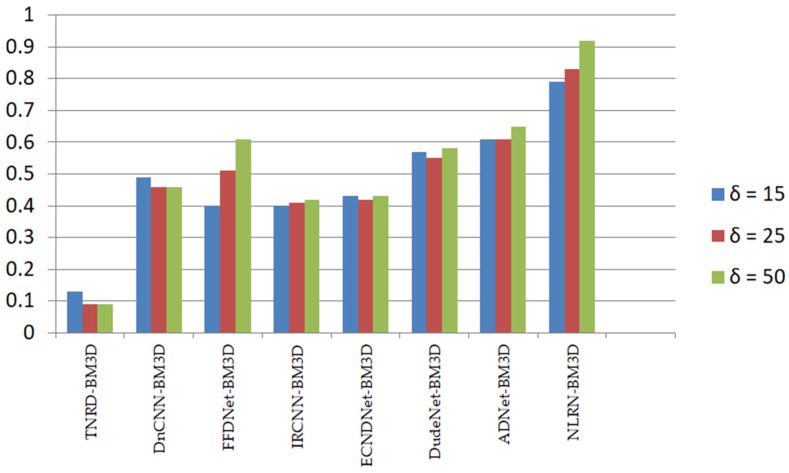
PSNR growth results of different methods relative to BM3D.

**Figure 3 sensors-25-02615-f003:**
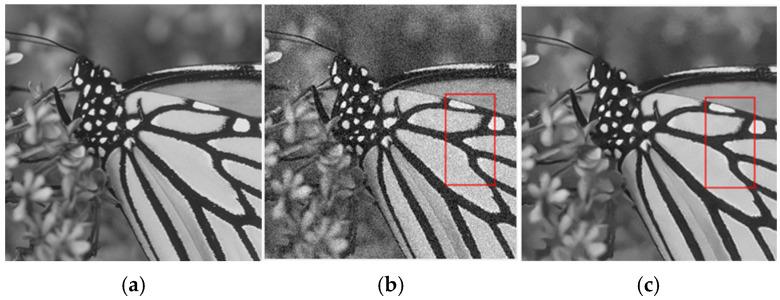
Denoising effect of different methods on Gaussian white noise image (δ = 15). (**a**) Original image; (**b**) noisy image; (**c**) NLM/30.32 dB; (**d**) BM3D/31.85 dB; (**e**) WNNM/32.71 dB; (**f**) EPLL/32.10 dB; (**g**) TNRD/32.27 dB; (**h**) DnCNN/33.09 dB; (**i**) DudeNet/32.93 dB; (**j**) FFDNet/32.66 dB; (**k**) IRCNN/32.82 dB; (**l**) NLRN/33.02 dB.

**Figure 4 sensors-25-02615-f004:**
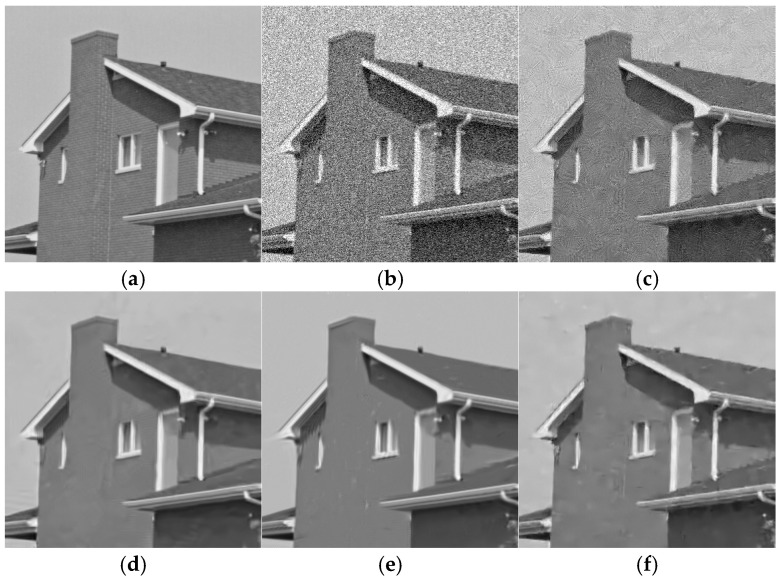
Denoising effect of different methods on Gaussian white noise image (δ = 25). (**a**) Original image; (**b**) noisy image; (**c**) NLM/26.72 dB; (**d**) BM3D/32.96 dB; (**e**) WNNM/33.21 dB; (**f**) EPLL/32.08 dB; (**g**) TNRD/32.94 dB; (**h**) DnCNN/33.08 dB; (**i**) DudeNet/33.02 dB; (**j**) FFDNet/33.27 dB; (**k**) IRCNN/33.07 dB; (**l**) NLRN/33.57 dB.

**Figure 5 sensors-25-02615-f005:**
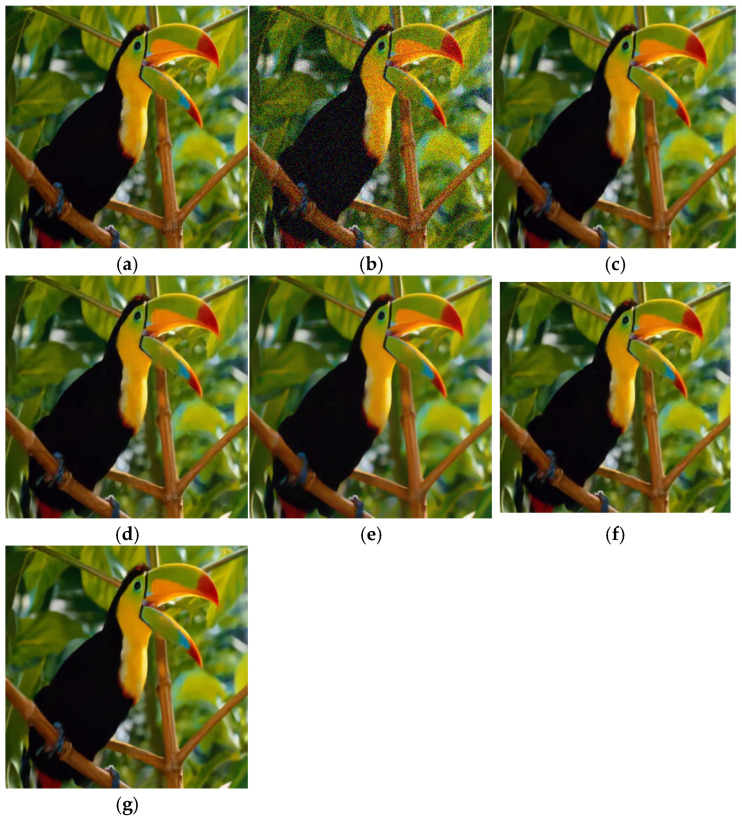
Denoising effect of different methods on Set5 (δ = 25). (**a**) Original image; (**b**) noisy image/20.18 dB; (**c**) CBM3D/32.38 dB; (**d**) DNCNN/32.39 dB; (**e**) FFDNet/32.88 dB; (**f**) IRCNN/32.56 dB; (**g**) BRDNet/33.12 dB.

**Figure 6 sensors-25-02615-f006:**
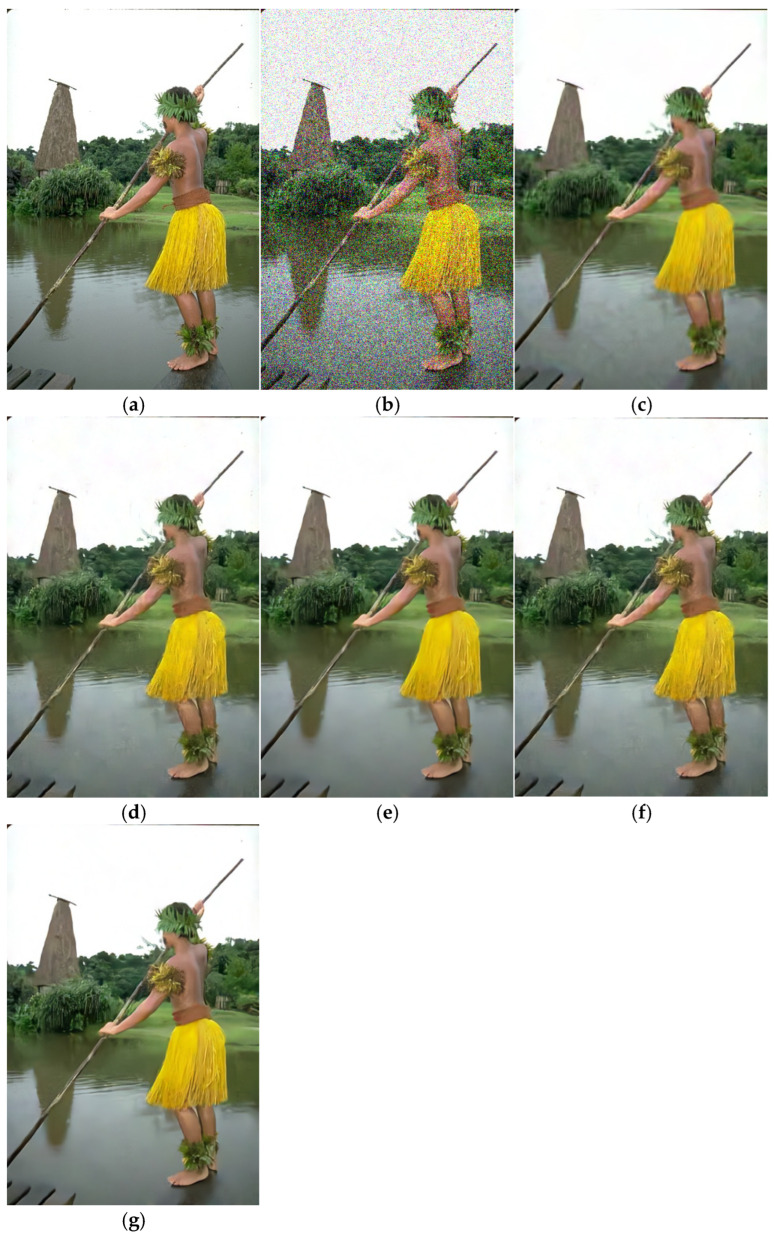
Denoising effect of different methods on CBSD68 (δ = 50). (**a**) Original image; (**b**) noisy image/14.15 dB; (**c**) CBM3D/28.32 dB; (**d**) DNCNN/28.93 dB; (**e**) FFDNet/28.93 dB; (**f**) IRCNN/29.02 dB; (**g**) BRDNet/29.39 dB.

**Figure 7 sensors-25-02615-f007:**
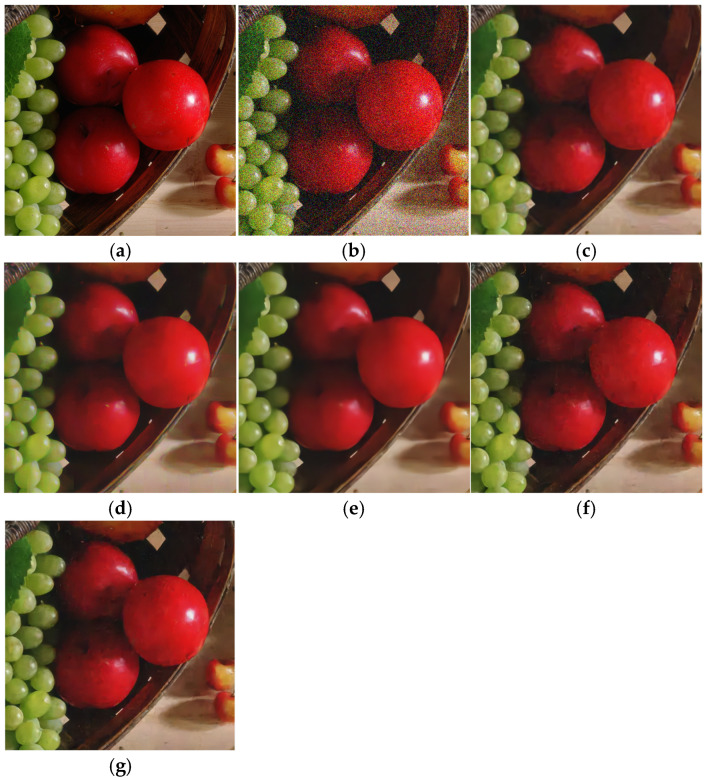
Denoising effect of different methods on McMaster (δ = 50). (**a**) Original image; (**b**) noisy image/14.16 dB; (**c**) CBM3D/30.94 dB; (**d**) DNCNN/31.17 dB; (**e**) FFDNet/31.65 dB; (**f**) IRCNN/31.39 dB; (**g**) BRDNet/32.31 dB.

**Table 1 sensors-25-02615-t001:** Traditional denoising approaches.

Type of Method	Method Name	Type of Noise	Descriptions
Filter	TVCT [34]	AWGN	Fully variable difference curve wave transform algorithm
	BFWIT [35]	Mixed noise	Bilateral filtering for hierarchical thresholding of wavelets
	AWT [36]	AWGN	Adaptive thresholding wavelet denoising
	BIDA [37]	AWGN	Enhanced image-denoising algorithm
	NLM [23]	AWGN	Nonlocal mean filtering
	BM3D [10]	AWGN	Sparse 3D transform-domain co-filtering
	CBM3D [11]	AWGN	BM3D YUV color mode conversion
Sparse coding	K-SVD [19]	AWGN	Sparse and redundant representation denoising methods
	NCSR [38]	AWGN	Nonlocalized concentration
	TWSC [39]	real noise	Three-sided weighted sparse coding
	LSSC [40]	Real, mixed noise	Nonlocal sparse model
	GDSR [41]	AWGN	Sparse representation denoising based on joint group dictionary
	GLRSTL [42]	AWGN	Graph Laplace regularized sparse transform learning
Model-based	EPLL [43]	AWGN	Model learning from natural image blocks
	NLH [44]	AWGN	Pixel-level nonlocal self-similarity
	NMOG [45]	AWGN	Low-rank matrix recovery methods
	WNNM [31]	AWGN	Weighted nuclear norm minimization
	GHP [46]	AWGN	Gradient histogram and texture enhancement
	LRTDTV [47]	AWGN	Low-rank tensor approach
	WCNNM [48]	AWGN	Multi-channel weighted nuclear specification minimization
	PLR [49]	AWGN	Fast image denoising based on patch with low-rank minimization
	PGPD [50]	AWGN	Nonlocal self-similarity based on patch groups
	FastHyde [51]	AWGN	Low-rank and sparse representation denoising methods
	GID [52]	AWGN	External data guidance and internal a priori learning

**Table 2 sensors-25-02615-t002:** Deep learning-based denoising approaches for additive white noise image.

Method	Technical Characteristics
MLP [59]	Multilayer perceptron.
TNRD [60]	An unfolded network structure based on nonlinear reaction–diffusion models; Combining traditional optimization methods with deep learning.
ECNDNet [61]	Residual learning and batch normalization techniques to address network training difficulties; Expanding the capture of contextual information using null convolution.
IDDRL [62]	A cascaded Gaussian conditional random field was used to optimize the model parameters through iterative learning; Deep residual learning network.
PSN-K [63]	A sparse representation and residual constraints were used to remove the noise.
DnCNN [55]	Residual learning and batch normalization.
ADNet [64]	Introducing an attention mechanism to dynamically assign feature weights.
BRDNet [65]	Combine two networks to increase network width; Combine two-way loop structure to capture image local and global dependencies.
NLRN [66]	Fusion of nonlocal similarity modeling with recurrent network stepwise refinement of denoising results.
DAGL [67]	Combination of graph convolutional networks and dynamic attention for effective modeling of nonlocal information in images.
DVA [68]	Unsupervised neural network and self-encoder architecture.
WAGFNet [69]	Wavelet anisotropic convolutional neural networks.
WINNet [70]	A two-branch architecture integrating wavelet transform and deep learning; Using reversible neural network (INN) architecture to avoid information loss during image sampling.
DeamNet [71]	Multiscale feature fusion and edge-aware design.
TripleDIP [72]	Deep Image Prior (DIP) framework; Three constraints (noise distribution, structural similarity, and sparsity) were introduced to optimize the generation process.
FEUSNet [73]	Fourier-embedded U-Net.
DPHSIR [74]	Combining pre-trained deep neural networks as a priori knowledge with classical optimization algorithms.
NHNet [75]	Using two sub-paths to process noisy images with different resolutions; Combining nonlocal operations to obtain effective features.

**Table 3 sensors-25-02615-t003:** Deep learning-based denoising approaches for realistic noisy images.

Method	Technical Characteristics
IRCNN [81]	Training a set of fast and efficient CNNs and integrating them into a model-based optimization approach; Accelerating training using batch normalization and residual learning.
DANet [82]	Removing real noise by the confrontation between the denoising network and noise-generating network.
MIRNet [83]	Parallel convolutional and multiscale fusion networks.
DNCNN [55]	Residual learning and batch normalization.
SSECNN [84]	Self-supervised structural similarity-based convolutional network.
Node2Node [85]	Self-supervised cardiac diffusion tensor.
RHUPL [86]	Robust hyperspectral unmixing with practical learning.
SUNet [87]	Transformer combined with U-Net improved ability to capture contextual information; Dual upsampling module prevented artifacts.
FEUSNet [73]	Fusion of Fourier features, based on their amplitude spectrum and phase spectrum properties; Embedding the learned features as a priori modules in a U-Net.
CDN [88]	Image information path (IIP) and noise estimation path (NEP) combined; Compositing information and self-similarity.
RIDNet [89]	Integrating multiscale feature extraction and attention mechanism.
MHCNN [90]	Multiple head convolutional neural network; A new multipath attention mechanism (MPA).

**Table 4 sensors-25-02615-t004:** Deep learning-based denoising approaches for blind noise image.

Method	Technical Characteristics
CBDNet [92]	The model was composed of a noise estimation subnetwork and a non-blind denoising subnetwork.
DIBD [94]	De-noising network based on degraded information learning.
DudeNet [95]	Using the dual-path network to increase the width of the network, to obtain more features; Using sparse mechanisms to extract global and local features
GCBD [96]	GAN-based blind denoiser.
SCNN [97]	The alternate direction multiplier (ADMM) method and the semi-quadratic segmentation method were used; Integrate the trained CNN noise-canceller into the model-based optimization method.
DNW [98]	Tunable convolutional neural network.
RDASNet [93]	Residual learning and dense connection and attention mechanism.
FFDNet [56]	Input noise level adjustable convolutional neural network.
GCDN [99]	Graph convolutional denoising network.

**Table 5 sensors-25-02615-t005:** Deep learning-based denoising approaches for mixed noise images.

Method	Technical Characteristics
SRMDNF [100]	Single convolutional super-resolution network for multiple degradations.
DnGAN [104]	Learning prior deep generative model.
TLCNN [105]	A new mapping from noisy to noise-free images using a four-stage CNN architecture; Adoption of transfer learning.
AINDNet + TF [106]	Adopting an adaptive instance normalization to build a denoiser
ICycleGAN [107]	Adding the EMA attention mechanism to the traditional residual module structure; Proposing a Resnet-E feature extraction module.

**Table 6 sensors-25-02615-t006:** Synthetic noise image datasets.

Category	Name	Color	Number	Size
Training dataset	BSD432	gray	432	481 × 321, 321 × 481
	CBSD432	gray	432	481 × 321, 321 × 481
	DIV2K	Gray, RGB	800	Sizes vary, about 1 k, 2 k
Test dataset	Set5	gray	5	280 × 280
	Set12	gray	12	256 × 256
	BSD68	gray	68	481 × 321, 321 × 481
	CBSD68	RGB	68	481 × 321, 321 × 481
	Kodak24	RGB	24	500 × 500
	McMaster	RGB	18	500 × 500

**Table 7 sensors-25-02615-t007:** Real noise image dataset.

Category	Name	Acquisition Mode	Camera	ISO
Test dataset	CC [114]	The third method	Canon 5D Mark	3.2 k
			Nikon D600	1.6 k
			Nikon D800	1.6 k, 3.2 k, 6.4 k
Test dataset	DND [115]	The second method	Sony A7R	100~25.6 k
			Olympus E-M10	200~25.6 k
			Sony RX100 IV	125~8 k
			Huawei Nexus 6P	100~6.4 k
Test dataset	SIDD [116]	The third method	Google Pixel	50~10 k
			iPhone 7	100~2 k
			Samsung Galaxy S6 Edge	100~3.2 k
			Motorola Nexus 6	100~3.2 k
			LG G4	100~800
Training dataset	Poly [117]	The third method	Canon 5D	3.2 k, 6.4 k
			Canon 80D	800~12.8 k
			Canon 600D	1.6 k, 3.2 k
			Nikon D800	1.6 k~6.4 k
			Sony A7	1.6 k, 3.2 k, 6.4 k

**Table 8 sensors-25-02615-t008:** Average PSNR (dB) for different approaches in Set12.

Method	δ=15	δ=25	δ=50
NLM	30.89	28.56	22.55
BM3D	32.37	29.97	26.72
WNNM	32.70	30.26	27.05
EPLL	32.14	29.69	26.47
MLP	—	30.03	26.78
TNRD	32.50	30.06	26.81
DnCNN	32.86	30.43	27.18
FFDNet	32.77	30.48	27.33
IRCNN	32.77	30.38	27.14
ECNDNet	32.80	30.39	27.15
DudeNet	32.94	30.52	27.30
ADNet	32.98	30.58	27.37
NLRN	**33.16**	**30.80**	**27.64**
RNAN	-	-	27.62

The bold one in the table is the best indicator.

**Table 9 sensors-25-02615-t009:** Average time for processing a single image using different algorithms (512 × 512).

Method	NLM	BM3D	WNNM	EPLL	TNRD	DnCNN	DudeNet	FFDNet	IRCNN	NLRN
Time(s)	230	2.26	740	42	1.96	3.12	7.19	1.16	6.31	4.17

**Table 10 sensors-25-02615-t010:** Average PSNR (dB) of different approaches on the color dataset.

Dataset	σ	CBM3D	DNCNN	FFDNET	DSNet	BRDNet	RPCNN	IRCNN
CBSD68	15	33.52	33.98	33.871	33.91	**34.10**	-	33.86
	25	30.71	31.31	31.21	31.28	**31.43**	31.24	31.16
	50	27.38	28.01	27.96	28.04	**28.16**	28 06	27.86
Kodak24	15	34.28	34.73	34.55	34.63	**34.88**	-	34.56
	25	31.68	32.23	32.11	32.l6	**32.41**	32 34	32.03
	50	28.46	29.02	28.99	29.05	29.22	**29.25**	28.81
Set5	15	34.04	34.29	34.31	34.17	**34.57**	-	34.26
	25	31.65	31.91	32.11	32.29	**32.51**	32.14	31.98
	50	28.69	28.96	29.22	29.06	**29.31**	29.04	29.00
McMaster	15	34.06	33.45	34.66	33.92	**35.08**	34.96	34.58
	25	31.66	31.52	32.35	32.14	**32.75**	32.34	32.18
	50	28.51	28.62	29.18	28.93	**29.52**	29.31	28.91

The bold one in the table is the best indicator.

**Table 11 sensors-25-02615-t011:** Average time for processing a single image using different algorithms (280 × 280).

Method	CBM3D	DNCNN	FFDNET	BRDNet	IRCNN
Time (s)	1.46	1.37	0.63	3.23	4.82

**Table 12 sensors-25-02615-t012:** PSNR and SSIM for different approaches on the two datasets.

Method	SIDD		DND	
	PSNR (dB)	SSIM	PSNR (dB)	SSIM
CBM3D	25.65	0.685	34.51	0.851
EPLL	27.11	0.870	33.51	0.824
K-SVD	26.88	0.842	36.49	0.899
DnCNN	32.59	0.861	37.90	0.943
FFDNet	38.27	0.948	37.61	0.942
CBDNet	33.28	0.868	38.06	0.942
RIDNet	38.71	0.951	39.26	0.953
VDNet [122]	39.26	0.955	39.38	0.952
GMSNet-A [123]	39.51	0.958	40.15	0.961
GMSNet-B [123]	39.69	0.958	**40.24**	**0.962**
MPRNet [124]	**39.71**	**0.958**	39.80	0.954

The bold one in the table is the best indicator.

**Table 13 sensors-25-02615-t013:** Average PSNR (dB) values of different algorithms on CC15 and PolyU datasets.

Dataset	CBM3D	NLH	DNCNN	FFDNet	MIRNet
CC15	37.95	**38.49**	37.47	37.68	36.06
PolyU	**38.81**	38.36	38.51	38.56	37.49

The bold one in the table is the best indicator.

## Data Availability

Data are contained within the article.

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
