# Peer review of "Overview of Research on Digital Image Denoising Methods"

_sensors, 2025, doi:10.3390/s25082615_

Round 1
Reviewer 1 Report
Comments and Suggestions for Authors
The paper's topic (Overview of Research on Digital Image Denoising Methods) is interesting and needs some revisions before publishing. Please address the following comments:
- Give further information on the dataset processing used to compare the denoising techniques.
- Before using denoising algorithms, describe any preparatory actions that were conducted.
- Some disadvantages of both conventional and deep learning-based denoising techniques are mentioned in the paper. A more thorough explanation of each strategy's drawbacks would be helpful.
- Although many deep learning models are compared in the study, it would be helpful to explain the selection of some architectures over others.
- More practical uses outside of datasets, such remote sensing or medical imaging, would enhance the conversation.
- Additional visual comparisons of denoised photos, such as side-by-side before/after photographs, could clarify the results, even though tables and figures are supplied.
- Given the substantial computing demands of deep learning models, it would be beneficial to examine the trade-offs between efficiency and performance.
- How were the deep learning models' hyperparameters selected? For every dataset, were they optimized?
- Did the research take into account any hybrid strategies that blend conventional and deep learning-based denoising techniques?
- When exposed to invisible noise types that aren't included in the training datasets, how do the models perform?
- Can the suggested deep learning techniques be successfully applied to noisy photos in the real world outside of the study's datasets?
- Have real-time applications like video denoising been used to evaluate the models?
- English needs some improvement.
Comments on the Quality of English Language
English needs some improvement.
Reviewer 2 Report
Comments and Suggestions for Authors
Image denoising techniques are an important research topic in image processing, and over the years, image denoising methods have been continuously improved. The authors analyzed the basic frameworks of classical traditional and deep neural network-based denoising methods and classified them according to their principles, characteristics, and other factors. Next, they carried out both quantitative and qualitative analyses. Finally, they pointed out the challenges and tried to look forward to the future research direction.
The authors mention many algorithms, although their short descriptions are sometimes too concise. For people who are not familiar with these methods, it is not enough to have a general idea about them. I think it would be worthwhile to describe in some detail more selected representatives of given classes of methods.
When describing data sets used in research, they should be cited in more detail. Please look into Deng, J.; Dong, W.; Socher, R.; Li, L.J.; Li, K.; Fei-Fei, L. Imagenet: A large-scale hierarchical image database. In Proceedings of the 2009 IEEE Conference on Computer Vision and Pattern Recognition, Miami, FL, USA, 20–25 June 2009; pp. 248–255. Section 3. should have a few sentences of introduction.
Many experiments were conducted, however, the course of the experiments was not properly and precisely described. It is not clear how the original images were noisy, how the parameters for the tested methods were selected, etc. It is worth adding an explanation as to why in some table columns, there are missing results for some methods.
The manuscript lacks a summary and an indication of the advantages and disadvantages of different methods. The results of the experiments should be described in detail. It would be worth analyzing what types of images specific methods are best suited for. When making comparisons, it is also worth taking into account the computational complexity of the algorithms used.
Some minor comments are below.
Section 2.2. should have subsections about additive Gaussian white noise images (line 189), real noisy images (line 219), blind noise images (line 235), mixed noise images (249).
Instead of citing the "literature [number]" it would be better to give the names of the authors, e.g. " In literature[9] an modified curvature filtering algorithm has been proposed" (line 113), "Literature[10] designed ..." (line 121), "Literature [11] accomplished ..." (line 127), etc.
It is also worth correcting minor editorial errors such as the size of letters, e.g.
" First, The essential framework" (line 21) "However, Images were" (line 35) or the spacing between words, e.g. "denoising models[2].AWNI mainly referred"(lines 45-46) "Gradient hi stogram estimation" (line 513). In line 88, "1. Introduction" should be deleted.
In line 193, " iiterature [46]" should be literature or better: Tai, Yang et al.
Reviewer 3 Report
Comments and Suggestions for Authors
- The title of the work does not properly reflect its content. You considered only several limited groups of denoising methods: two dimensional amplitude-type images (while there are also phase images, and different kinds of the-dimensional images: hyperspectral, spatio-temporal, tomographic, terahertz). The easiest way is to correct the title, while mentioning all image types, and providing the references on denoising algorithms for such image types is able to enrich the review. I recommend both actions. To learn more about these different type of denoising methods (hyperspectral, spatio-temporal, tomographic, terahertz) I can recommend considering the last ten year publications of Vladimir Katkovnik.
- The results of analysis provided are quite limited. Please consider the latest reviews (e.g. by Silvio Montresor, Pietro Ferraro group [https://doi.org/10.1038/lsa.2016.142, https://doi.org/10.1038/s41377-018-0050-9]) and add more comparison cases, etc.
- Low writing quality. E.g. the first two sentences of the introduction seem very strange in combination with the third one.
- The Figures are also inconsistent: all math symbols in the doggies and in the text should be written with italic font. Fig. 1 is very primitive, while now the complex nature of the nose is not discussed in the paper. When the nose is Gaussian [https://doi.org/10.1016/j.sigpro.2018.12.006], when is it Poissonian [https://doi.org/10.1016/j.bspc.2024.106207], when it can be described by Skellam distribution [https://dx.doi.org/10.1364/AO.58.000G61]? the difference with light (emitter) source noise and detector noise should be discussed and visualized. Please consider all these details in the revised version.
- The text on Figure 2 should be rotated on the 180 degrees, to read it in a printed booklet version. Why do stages almost duplicate each other? The original BM3D scheme is different.
- Figure 3 is also very common and because of that is useless. To show the qualification to be worth publishing a review paper, you need to analyze all the discussed algorithms of such family and provide a generalized diagram of all these methods, at the same time reflecting their unique features in small but very important details. Otherwise, what does your work give to the reader?
- You provided more than 100 references, but the methods just tested on one dataset. Please check the Karen Egiazaryan papers to find more datasets and metrics. Moreover, different methods have different prerequisites for the development, so some of them should be effective in other conditions. Your current version of the work has failed to reflect this fact in full details.
- The conclusion is not to provides the summary for the work (does not highlight unique findings obtained in the work), while it just describe again what has been done.
Comments on the Quality of English Language
- The manuscript contains the misprints even in the abstract: "First, The..." (Two Capital letters), and should be programmed carefully. Such and other problems are found further ("However, Images", etc.).
- The sentence in the abstract, where the abbreviation BM3D is introduced contains errors. At first, the abbreviation is wrongly decrypted. Instead of "matched", the words "matching" should be used. At second, it is not written in accordance with the style rules. You mentioned that the algorithm removes noise twice, in the beginning and at the end of the sentence.
- There are a lot of problems with English in this document. E.g. grayscale is written with "e". Even standard spellchecker has not been used. Another example of careless attitude to the journal and reviewers is at the line 88: "work.1. Introduction". What does it mean? Or lines 90-91: "noise mainly came from image acquisition,... such as acquisition process...." Instead of providing an example here you just repeated the same. Phrases like "Literature [12]" also should be corrected. So many basic problems, that the manuscript deserves rejection only for this and resubmission after proofreading with the professional services. More examples: "In 2018 Literature [18] recommended..." If this is not review work, the word recommended is inappropriate there. The reference list was also formatted with errors. E.g. the title of the work in Ref. [48] contains words "cnn", while such abbreviations should be written as "CNN".
- The text contains unprofessional sentences, like: "after denoising process, the output image as clear as possible". At first, the verb "is" is missed there. Secondly, what if the denoising process was unsuccessful? By the way, the next sentence after that is also similar to this in terms of lack of professional proofreading.
Round 2
Reviewer 1 Report
Comments and Suggestions for Authors
Accept in present form.
Reviewer 3 Report
Comments and Suggestions for Authors
Thank you for the quite details revision of the work. I believe, that the quality of the work increased.